# Trend of home birth and its associated factors in Ethiopia during COVID-19 and social crisis (2019–2023)

**Fekade Demeke Bayou**📷[1]*, **Fekadeselassie Belege Getaneh**[2], **Lakew Asmare**[3], **Abel Endawkie**[1], **Alemu Gedefie**[4], **Amare Muche**📷[1], **Anissa Mohammed**[1], **Aznamariam Ayres**📷[1], **Dagnachew Melak**[1], **Eyob Tilahun Abeje**[1]

1 Department of Epidemiology and Biostatistics, School of Public Health, College of Medicine and Health Sciences, Wollo University, Dessie, Ethiopia, 2 Department of Pediatrics and Child Health Nursing, College of Medicine and Health Sciences, Wollo University, Dessie, Ethiopia, 3 Department of Epidemiology and Biostatistics, Institute of Public Health, College of Medicine and Health Science, University of Gondar, Gondar, Ethiopia, 4 Department of Medical Laboratory Sciences, College of Medicine and Health Sciences, Wollo University, Dessie, Ethiopia

* fekadedemeke12@gmail.com

## Abstract

### Background

Maternal mortality is unacceptably high in some countries of the world, including Ethiopia. Access to skilled delivery is one of the prevention methods for maternal and neonatal deaths. However, a significant number of women gave birth at home due to many reasons. In, Ethiopia, after the implementation of many interventions to reduce home birth, the change in its trend was not studied. Hence, this study was aimed to address this information gap.

### Objective

To determine the trend of home birth and its associated factors in Ethiopia during COVID-19 and Social Crisis from 2019–2023.

### Method

We obtained the data from the Performance Monitoring for Action, which employed panel design with embedded cross-sectional surveys from 2019–2023. A total sample of 8,419 women who gave birth were included in this analysis. A two-stage cluster sampling was applied to select study participants. The datasets were appended, edited, and analyzed using STATA 17 and SPSS 25. Microsoft Excel was used to prepare line graphs. A binary logistic regression model was fitted to identify factors associated with home delivery. We used an adjusted odds ratio with 95% CI to measure the strength of associations.

**Data availability statement:** The dataset used in this study is owned by third party organization. Interested body can communicate them for the dataset via: https://www.pmadata.org. Users are required to register for a PMA account to request access to the datasets. Access to the datasets are granted on a per request basis. Once approved, users can download and use the data for free. To request datasets, first create an account or login to an existing account. Then submit request for the datasets of interest (S1 File).

**Funding:** The author(s) received no specific funding for this work.

**Competing interests:** The authors have declared that no competing interest exist.

**Abbreviations:** ANC, Antenatal Care; LMIC, Lower- and Middle-Income Countries; PMA, Performance Monitoring for Action; SDG, Sustainable Development Goals

## Result

The level of home birth showed a significant decline from 37.80% in 2019 to 29.90% in 2023. Having less than four ANC visits [AOR = 1.91, 95% CI (1.25, 2.92)], not encouraged by partners to visit clinics [AOR = 2.15, 95%CI (1.19, 3.87)], and did not discuss delivery by skilled attendant during ANC visit [AOR = 1.70, 95% CI (1.09, 2.64)] were some of the variables significantly associated with home birth.

## Conclusion

Home birth is declining in Ethiopia. Providers and health managers should use ANC visits as a good opportunity to inform, encourage and follow mothers to attend birth at a health facility.

## Introduction

Maternal mortality is unacceptably high with almost 800 women dying every day in 2020 mainly from preventable causes [1,2]. In 2020, nearly 95% of maternal deaths occurred in low and lower-middle-income countries, Sub-Saharan Africa alone accounted for around 70% (202, 000) of maternal deaths [3]. In Ethiopia, in 2022, the maternal mortality ratio was 401 per 100,000 live births [4]. Most of the deaths were attributable to severe bleeding, infections, hypertension, complications during and after giving birth, and unsafe abortion [5].

The high number of maternal deaths in some areas of the world reflects inequalities in access to quality health services and highlights the gap between rich and poor [6]. Humanitarian, conflict, and post-conflict settings hinder progress in reducing the burden of maternal mortality [7,8]. Most maternal deaths are preventable, as the healthcare solutions to prevent or manage complications are well known [3]. Improving institutional birth is one of the strategies to improve maternal health in lower and middle-income countries (LMIC) [9]. Across the world, Sustainable Development Goal (SDG) 3.1 targeted to achieve a maternal mortality ratio of less than 70 per 100,000 live births by 2030 [10]. Ethiopia is striving to achieve the SDG 3.1 target, however, the high prevalence (73.0%) of home birth preference is one of the major challenges to reducing maternal mortality in the country [11,12]. In Ethiopia, a study based on demographic and health survey (DHS) data revealed that the trend of home birth showed a significant decline from 94.20% in 2005 to 73.44% in 2016 [13]. Previous studies conducted in different parts of Ethiopia found a high prevalence of home birth (54.50%–80.00%) [14–17]. This figure showed the country is behind the WHO recommendation, that all women should have access to skilled care during pregnancy and childbirth to ensure prevention, detection, and management of complications [18,19].

In Ethiopia, home births are commonly attended by traditional birth attendants, no one and family or friends, who are unskilled and unable to prevent or treat the complications during pregnancy or childbirth that lead to maternal deaths [20–22]. Evidence shows that women who give birth at home are supposed to experience adverse birth outcomes including postpartum hemorrhage (50.60%), retained placenta/placental tissues (28.40%), traumatic lesions (ruptured uterus, second- degree perianal tear) (18.40%), and puerperal sepsis (10.70%) [23].

Home birth preferences might arise from pregnant women, family/household or health system-related factors. Pregnant women with no formal education, poor knowledge of obstetric complications, negative attitudes towards birth services, no birth preparedness, no media access, no antenatal care visit, and discussion about the place of birth were more

likely to give birth at home [11,24–26]. Moreover, cultural beliefs and practices like opportunities to access psychological support through family members, culturally acceptable food, a birthing position of choice, and safe culturally accepted disposal of placenta and misperceptions and worries on medical interventions are highly influencing preference of home births [27,28]. Household or family factors including poor household wealth index or low monthly income [24], husbands being decision-makers [26], rural residence, and distance from facilities were found to be potential determinants of home birth [11,25,29]. Moreover, poor service delivery at the facility, poor competence of providers, and limited availability of supplies and equipment were found to maintain the preference to give birth at home [30].

As part of maternal mortality prevention, recently the Ethiopian government has planned to increase births attended by skilled health personnel (doctors, nurses/midwives, health officers, and health extension workers) from 50% (in 2019) to 76% (in 2024/25) [22,31]. On the other hand, the country faced many challenges including COVID-19 and social conflicts which would impede the usual healthcare system. After such efforts and challenges, the level of home birth has not been assessed. Hence, this study aimed at assessing the change in the level of home birth from 2019 to 2023 and identifying its determinant factors in Ethiopia.

Ethiopia is the second most populous country in Africa, with approximately 107 million population in 2023, a rapid growth rate (2.6%), and a predominantly young age structure. The country has a high total fertility rate (4.6 births per woman) and an estimated crude birth rate of 27 per 1,000 [32].

## Methods

### Study setting

The study was conducted in predominantly agrarian regions of Ethiopia including Afar, Tigray, Oromia, Amhara, and Southern Nations, Nationalities, and Peoples' Region, and one urban region, Addis Ababa.

### Study design and data source

Performance Monitoring for Action (PMA) employed panel design with embedded cross-sectional surveys. We obtained the data for this study from four recent rounds of PMA surveys conducted from 2019–2023. The data from PMA surveys were accessed from the repository after a formal request by the principal investigator (FDB) (https://www.pmadata.org/data/request-access-datasets) (S1 File). PMA Ethiopia was a five-year project being implemented since 2014 (formerly as Performance Monitoring and Accountability 2020 (PMA2020)). The PMA surveys were repeated cross-sectional surveys addressing health issues including key reproductive, maternal and newborn health (RMNH) indicators. Survey implementation was managed by Addis Ababa University, School of Public Health (AAU) in collaboration with regional universities, the Federal Ministry of Health, and the Central Statistics Agency. Technical support was provided by the Bill and Melinda Gates Institute for Population and Reproductive Health at the Johns Hopkins Bloomberg School of Public Health. The project was funded by the Bill & Melinda Gates Foundation and managed by the Ethiopian Public Health Association (EPHA) [33].

### Study participants

Post-partum women aged 15–49 years residing in households, located within the survey regions and with complete information were included in this study.

## Sample and sampling technique

A two-stage cluster sampling with urban-rural, and major regions as strata was applied to select study participants. First, areas located in the sentinel regions were stratified into urban and rural. Then, enumeration areas (EAs) were randomly selected from the master sample frame of the central statistical agency [34]. Second, households were randomly selected from within each enumeration area. All post-partum women aged 15–49 years with complete information were included for the analysis in this study.

## Data collection

The PMA surveys are designed to collect nationally or sub-nationally representative data on key women's health measures through three related data collection activities: Household and Female surveys (HQFQ), Service Delivery Point (SDP) surveys, and Service Delivery Point Client Exit Interview (CEI) surveys. For our study, we used data collected by Household and Female surveys (HQFQ), which includes measures on awareness, perception, knowledge, and use of contraceptive methods, components of health service provision, perceived quality, and side effects of method among current users, birth history, fertility intentions, and domains of empowerment. The household questionnaire collects basic household roster information and information on observed dwelling unit characteristics, such as type of floor and roof, which is used to generate a household wealth index.

## Outcome measurement

The outcome variable, place of birth (home or health facility birth) was assessed by using an interviewer-administered questionnaire. The questionnaire included: Where did you give birth? Probing questions to identify the type of facility. Women who gave birth at their own or others' homes were considered home birth. Whereas those who gave birth at health at private, governmental, or non-governmental health facilities were categorized as health facility birth.

## Data processing and analysis

The datasets containing four-year trends in home birth of six Ethiopian regions were obtained from the PMA data repository. Then the datasets were appended, cleaned, edited, recoded, and analyzed using STATA 17 and SPSS 25. We used descriptive statistics to present the trend of home births over time between 2019 and 2023. Microsoft Excel was used to prepare line graphs. A logistic regression model was fitted to identify factors associated with home birth. The model fitness was checked using the Hosmer and Lemeshow goodness of fitness test. Bivariable logistic regression was run to identify potential variables affecting place of birth. Predictor variables with $P < 0.25$ were used to fit multivariable logistic regression (final model). We used an adjusted odds ratio with 95% CI to measure the strength of associations.

## Ethics approval and consent to participate

Permission was obtained to use the PMA Data for the research purpose. Since we used secondary data, consent from each study participant was not applicable.

# Result

## Characteristics of study participants

We analyzed data for sample sizes ranged from 1694 to 2563 women. For two regions (namely Tigray and Afar, two years (2022 and 2023), were missing (Table 1).

**Table 1. Description of the study participants by regions and survey years in Ethiopia from 2019–2023.**

| Regions | Survey years | | | | Total |
|---|---|---|---|---|---|
| | **2019** | **2020** | **2022** | **2023** | |
| | Number of participants, N (%) | | | | |
| Tigray | 435 (17.0%) | 396 (17.3%) | – | – | 831 |
| Afar | 210 (8.2%) | 190 (8.3%) | – | – | 400 |
| Amhara | 448 (17.5%) | 403 (17.6% | 382 (22.6%) | 409 (21.8%) | 1642 |
| Oromia | 636 (24.8%) | 553 (24.2%) | 609 (36.0%) | 660 (35.2%) | 2458 |
| SNNP | 588 (22.9%) | 536 (23.4%) | 467 (27.6%) | 536 (28.6%) | 2127 |
| Addis Ababa | 246 (9.6%) | 208 (9.1%) | 236 (13.9%) | 271 (14.4%) | 961 |
| **Total** | 2563 (100.0%) | 2286 (100.0%) | 1694 (100.0%) | 1876 (100.0%) | 8419 |

Source of data: PMA Ethiopia.

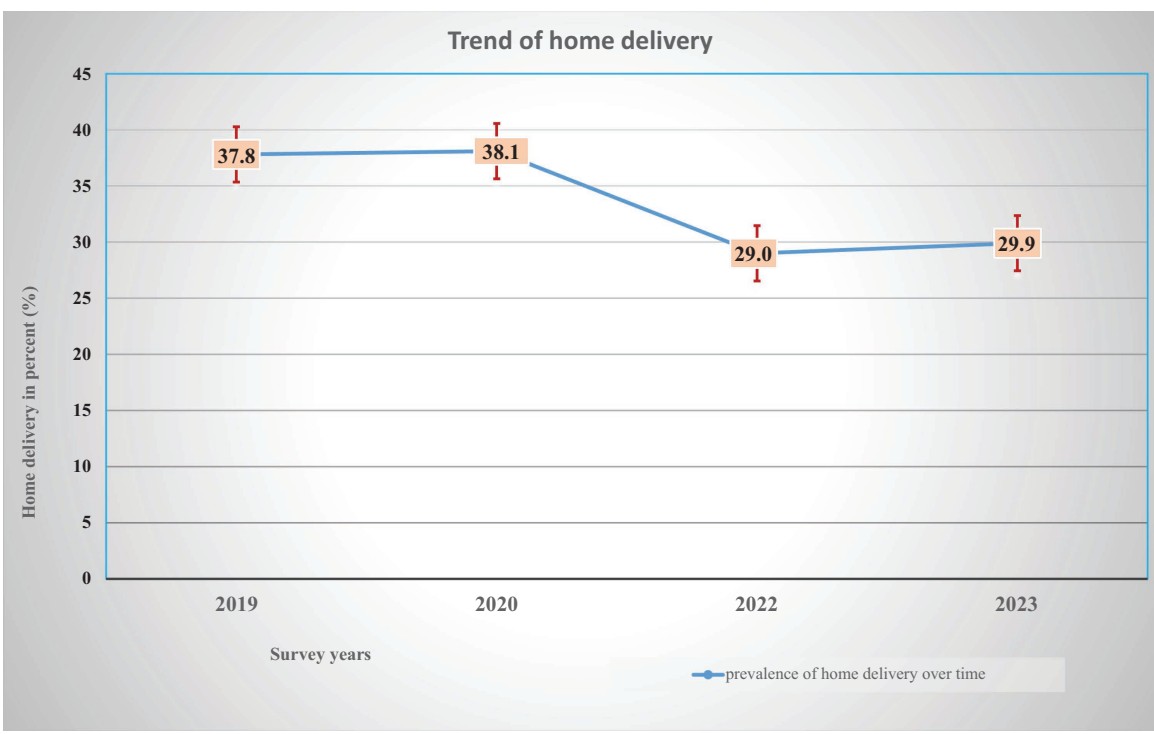

**Fig 1. Trend of home birth from 2019–2023 in Ethiopia (source of data: MA Ethiopia).**

## Trends of home birth

The trend of home birth over the study period (2019–2023) showed a significant decline, which is decreased from 37.80% in 2019 to 29.90% in 2023 (Fig 1).

We also computed changes in the proportion of home births from 2019 to 2020, from 2020 to 2022, from 2019 to 2022, from 2022 to 2023, and from 2019 to 2023. The largest decline in the proportion of home birth was observed from 2020 to 2022 with a 9.10% drop down followed by an 8.80% reduction from 2019 to 2022 (Table 2).

**Table 2. Change in the prevalence of home birth from 2019–2023 in Ethiopia.**

| Regions | 2019–2020 | 2020–2022 | 2019–2022 | 2022–2023 | 2019–2023 |
|---|---|---|---|---|---|
| | Differences in the proportion of home birth | | | | |
| Tigray | 0.6 | – | – | – | – |
| Afar | 0.8 | – | – | – | – |
| Amhara | −0.2 | −13.4 | −13.6 | 1.9 | −11.7 |
| Oromia | −0.5 | −5.9 | −6.4 | 1.7 | −4.7 |
| SNNP | 0.9 | −10.1 | −9.2 | −0.2 | −9.4 |
| Addis Ababa | −0.1 | −1.1 | −1.2 | −0.1 | −1.3 |
| Overall (all regions) | 0.3 | −9.1 | −8.8 | 0.9 | −7.9 |

Source of data: PMA Ethiopia.

### Factors associated with home birth

Factors associated with home birth were identified using multivariable logistic regression analysis. Accordingly, number of antenatal care (ANC) visits (having less than four ANC visits), not encouraged by partners to visit clinics, had not discussed about transport for delivery during ANC visits, did not discuss birth by a skilled attendant during ANC visits, and did not discuss their place of birth with partners were significantly associated with increased home birth. Women who visited health facilities less than four times during ANC were nearly two times more likely to give birth at home compared with those who visited four or more times [AOR = 1.91, 95%CI (1.25, 2.92)]. Similarly, women who were not encouraged by their partner to go to the clinic were more than two times more likely to give birth at home than their counterparts [AOR = 2.15, 95%CI (1.19, 3.87)]. Home birth was 1.7 times more likely among women who did not discuss birth by a skilled attendant during ANC visits than their counterparts [AOR = 1.70, 95%CI (1.09, 2.64)]. Moreover, women who did not discuss the place of birth with their partners were 2.5 times more likely to give birth at home [AOR = 2.48, 95%CI (1.53, 4.03)] (Table 3).

## Discussion

This study aimed at investigating the trend of home birth and its predictors in Ethiopia during the COVID-19 pandemic and social crisis era (2019–2023). The proportion of home births showed a significant decline in Ethiopia during the study period. Antenatal care (ANC) attendance, partner's encouragement, and discussion on the place of birth are the potential to determine home birth.

Trend of home birth reduced from 37.80% in 2019 to 29.90% in 2023 in Ethiopia. The current level of home birth is comparable with the estimated prevalence in low-middle-income countries (LMIC), which was 28.00% [12]. However, there is a time gap between the latter study and the currents, since the above study pooled evidence from 2000–2019, the figure might be somewhat outdated. While our finding indicated the most recent figure (2019–2023). This figure is much lower as compared to the pooled estimate of home birth among the pastoralist community in Ethiopia, which is 78.80% [35]. This might be because pastoralists experience barriers due to geographical location, sociocultural dynamics, availability of logistics, norm and traditions, and individual perception. This made nomad pastoralist women experience completely different from the rest of the world [36]. Moreover, differences in study time may be the cause for the observed discrepancy, in recent times, the establishment of maternity waiting homes (MWH) helps to improve access to skilled care by bringing pregnant women physically present or close to health facilities [37], and improvement of Health Extension Program to provide health education and in communities that promote healthcare services utilization during pregnancy and childbirth, which in turn decrease home childbirth.

**Table 3. Multivariable logistic regression analysis of the factors associated with home birth in Ethiopia from 2019–2023.**

| Variables | Categories | Place of Birth | | COR (95%CI) | AOR (95%CI) |
|---|---|---|---|---|---|
| | | Home Birth N (%) | Health facility N (%) | | |
| Number of ANC visits | Less than four | 221 (44.90) | 271 (55.10) | 2.51 (1.69, 3.72) | 1.91 (1.25, 2.92)* |
| | Four or more | 41 (24.60) | 126 (75.40) | 1 | 1 |
| Partner encouraged you to go clinic for antenatal care | No | 373 (76.00) | 118 (24.00) | 8.21 (6.51,10.35) | 2.15 (1.19, 3.87) * |
| | Yes | 499 (27.80) | 1296 (72.20) | 1 | 1 |
| Discussed about transport for delivery during ANC visit | No | 648 (54.30) | 546 (45.70) | 4.60 (3.82, 5.54) | 1.72 (1.16, 2.54) * |
| | Yes | 224 (20.50) | 868 (79.50) | 1 | 1 |
| Discussed delivery by skilled attendant during ANC visit | No | 562 (52.90) | 500 (47.10) | 3.31 (2.78, 3.95) | 1.70 (1.09, 2.64) * |
| | Yes | 310 (25.30) | 914 (74.70) | 1 | 1 |
| Experienced difficulty in accessing care for complications | No | 861 (38.10) | 1399 (61.9) | 1.19 (0.55, 2.61) | 0.42 (0.10, 1.81) |
| | Yes | 11 (42.30) | 15 (57.70) | 1 | 1 |
| Experienced convulsion during pregnancy | No | 804 (37.50) | 1339 (62.50) | 1 | 1 |
| | Yes | 68 (47.60) | 75 (52.40) | 1.51 (1.08, 2.12) | 0.57 (0.30, 1.10) |
| Experienced high Blood Pressure | No | 861 (38.60) | 1369 (61.40) | 1 | 1 |
| | Yes | 11 (19.60) | 45 (80.40) | 0.39 (0.20, 0.76) | 2.73 (0.40, 18.72) |
| Experienced Edema during pregnancy | No | 767 (40.10) | 1144 (59.90) | 1 | 1 |
| | Yes | 105 (28.00) | 270 (72.00) | 0.58 (0.46, 0.74) | 1.35 (0.82, 2.23) |
| Experienced vaginal bleeding during pregnancy | No | 854 (38.30) | 1374 (61.70) | 1 | 1 |
| | Yes | 18 (31.00) | 40 (69.00) | 0.72 (0.41, 1.27) | 0.33 (0.09, 1.28) |
| Covid-19 affected where you delivered? | No | 847 (38.50) | 1352 (61.50) | 1.55 (0.97, 2.49) | 0.94 (0.36, 2.44) |
| | Yes | 25 (28.70) | 62 (71.30) | 1 | 1 |
| Discussed where to deliver with partner | No | 481 (70.70) | 199 (29.30) | 7.51 (6.15, 9.18) | 2.48 (1.53, 4.03) * |
| | Yes | 391 (24.30) | 1215 (75.70) | 1 | 1 |

Source of data: PMA Ethiopia.

*Significant association ANC: Antenatal Care.

The current study found that women with three or fewer ANC visits were more likely to give birth at home as compared to their counterparts. This finding is supported by the study conducted by Ayenew et al 2021 [29]. The possible explanation for the observed association might be women who attended no or fewer ANC follow-ups could miss the opportunity to get information on dangerous signs of pregnancy, the danger of giving childbirth at home, birth preparedness and complication readiness, discussion about transport for childbirth during ANC visit, discussion about birth by a skilled attendant during ANC visit and discussion on the place of birth with a partner. As a result, they would be out of providers' close follow-up and encouragement to visit clinics, which leads them to give birth at home.

Our study revealed that lack of partners' encouragement to visit clinics, lack of discussion about transport, place of birth, and birth by skilled attendant during ANC visit were significantly associated with increased home birth. Discussion on birth plan and place of birth are among core activities during ANC visit. However, such crucial components are usually missing in many Sub-Saharan African countries. Supporting the findings by previous studies conducted elsewhere, the current study revealed that having no discussion regarding place of birth and transport issues are deterring factors for health facility birth [11,38]. This is because discussion with providers is the major source of pregnancy and related knowledge

for mothers, especially in less educated rural communities. Moreover, partners' involvement decision making regarding place of birth and encouragement of their wives to give birth at health facilities is paramount in countries like Ethiopia, where males are usually heads of the households and sources of income [15].

## Strength and limitation of the study

As a strength, this study relied on a series of data on place of birth in Ethiopia to investigate the time trend of home birth. This might be helpful to understand the past experiences, the status and future progression in terms of place of birth. This study relied on the data collected from a sampled region of Ethiopia. However, the level of home births have significant difference across regions [13]. Moreover, for the two regions (namely Tigray and Afar) the two years (2022 and 2023) data were missing. As a result, this finding might have under or overestimated the proportion of home birth in the country. Hence, the figures should be interpreted with caution and considering this limitation.

## Conclusion and recommendation

The current study revealed a significant reduction in home birth compared to the previous trends in Ethiopia. This is highly inspiring and a good indicator of the country's commitment to achieving the SDG target 3.1 (reducing maternal mortality to less than 70 per 100,000). Based on our finding, improving ANC visits, consultation on place of birth, the importance of skilled birth and involving partners to encourage their wives may help to further reduce home birth. Hence, providers and health managers should use ANC visits as a good opportunity to be aware, encourage and follow mothers to attend health facility birth.

## Supporting information

**S1 File. Searching strategy.**
(DOCX)

## Author contributions

**Conceptualization:** Fekade Demeke Bayou.

**Data curation:** Fekade Demeke Bayou, Fekadeselassie Belege Getaneh.

**Formal analysis:** Fekade Demeke Bayou, Fekadeselassie Belege Getaneh, Lakew Asmare, Abel Endawkie, Alemu Gedefie, Amare Muche, Anissa Mohammed, Aznamariam Ayres, Dagnachew Melak, Eyob Tilahun Abeje.

**Funding acquisition:** Fekade Demeke Bayou.

**Investigation:** Fekade Demeke Bayou.

**Methodology:** Fekade Demeke Bayou.

**Project administration:** Fekade Demeke Bayou.

**Resources:** Fekade Demeke Bayou.

**Software:** Fekade Demeke Bayou.

**Supervision:** Fekadeselassie Belege Getaneh, Lakew Asmare, Abel Endawkie, Alemu Gedefie, Amare Muche, Anissa Mohammed, Aznamariam Ayres, Dagnachew Melak, Eyob Tilahun Abeje.

**Validation:** Fekade Demeke Bayou, Fekadeselassie Belege Getaneh, Lakew Asmare, Abel Endawkie, Alemu Gedefie, Amare Muche, Anissa Mohammed, Aznamariam Ayres, Dagnachew Melak, Eyob Tilahun Abeje.

**Visualization:** Fekade Demeke Bayou, Fekadeselassie Belege Getaneh, Lakew Asmare, Abel Endawkie, Alemu Gedefie, Amare Muche, Anissa Mohammed, Aznamariam Ayres, Dagnachew Melak, Eyob Tilahun Abeje.

**Writing – original draft:** Fekade Demeke Bayou, Fekadeselassie Belege Getaneh, Lakew Asmare, Abel Endawkie, Alemu Gedefie, Amare Muche, Anissa Mohammed, Aznamariam Ayres, Dagnachew Melak, Eyob Tilahun Abeje.

**Writing – review & editing:** Fekade Demeke Bayou, Fekadeselassie Belege Getaneh, Lakew Asmare, Abel Endawkie, Alemu Gedefie, Amare Muche, Anissa Mohammed, Aznamariam Ayres, Dagnachew Melak, Eyob Tilahun Abeje.

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
