## [Decision Letter · Decision Letter 0]

1 Aug 2024

PONE-D-24-10835Trend of Home Delivery and Its Associated Factors in Ethiopia during COVID 19 and Social Crisis (2019-2023)PLOS ONE

Dear Dr. Bayou,

Thank you for submitting your manuscript to PLOS ONE. After careful consideration, we feel that it has merit but does not fully meet PLOS ONE’s publication criteria as it currently stands. Therefore, we invite you to submit a revised version of the manuscript that addresses the points raised during the review process.

**ACADEMIC EDITOR:**

The manuscript has very valuable contributions but it requires solving some flaws: editing tables and figures, references and some writing errors.I suggest that the authors strengthen consistency between the different sections: for example the role of personnel who attend home births versus hospital births.The reviewers agree on the need to unify the way of referring to home birth: birth or deliveryI considered that the major revision is the relevant decision to improve the manuscript due to inconsistencies that I point out below.

Please submit your revised manuscript by Sep 15 2024 11:59PM.  If you will need more time than this to complete your revisions, please reply to this message or contact the journal office at plosone@plos.org . Please include the following items when submitting your revised manuscript:

We look forward to receiving your revised manuscript.

Kind regards,

Doris Verónica Ortega-Altamirano, PhD

Academic Editor

PLOS ONE

file:///home/administrator/Downloads/Phase%202_3%20Protocol_Final_02_07_EN.pdf

https://pubmed.ncbi.nlm.nih.gov/34913391/

In your revision ensure you cite all your sources (including your own works), and quote or rephrase any duplicated text outside the methods section. Further consideration is dependent on these concerns being addressed.

3. In the online submission form you indicate that your data is not available for proprietary reasons and have provided a contact point for accessing this data. Please note that your current contact point is a co-author on this manuscript. According to our Data Policy, the contact point must not be an author on the manuscript and must be an institutional contact, ideally not an individual. Please revise your data statement to a non-author institutional point of contact, such as a data access or ethics committee, and send this to us via return email. Please also include contact information for the third party organization, and please include the full citation of where the data can be found.

Additional Editor Comments:

Abstract: the surveys analysed correspond to the period 2021-2023. However, the study period was from 2019 to 2023. Clarify this inconsistency.

Line 70-71”… substantial prevalence of home birth preference in Ethiopia…” The ratio of births at home to hospital is missing. How much substantial?

Line 74. Improve drafting. Clarify: Prevalence of which country Ethiopia or other African countries? The baseline should be reviewed to determine where the prevalence of 94.20% is from.

Lack information about the association, if there is between home-assisted delivery and the prevalence of maternal morbidity and mortality due to bleeding, infection and high blood pressure. Also clarify the fraction attributable to attention to birth given by attendee without sufficient training.

Line 81-86. The authors list different causes of preference for home births attributable to pregnant women and their family members and access to health services. Please clarify the idea and put separately the causes attributable to pregnant women and home birth, as well as the causes attributable to the health system, And human resources with insufficient training for childbirth care.

Line 92: Check if the idea is correct. If the number of home births was not measured, why are you planning to expand services from 50% to 76%?

Study Design and Data Source. Line 106-107: “We obtained the data for this study from four recent rounds of the Performance Monitoring for Action (PMA) surveys conducted from 2021-2023." The date of the data source 2021-2023 does not coincide with the study period mentioned in the manuscript title: 2019-2023. Where the data for the years 2019 and 2020 come from.

Line 109-112. Authors should specify in detail the agency responsible for PMA surveys.

Line 121-122. Reference of the sample frame missing: “Enumeration areas (EAs) were selected from the master sample frame of the Central Statistical Agency”.

Line 128-129: It is not clear how household surveys are included in the PMA.

Review text organization, for example:

Line 100-104. Information to include in the introduction.

Line 137-138: Information to be in the data source section.

Line 147. “Four years trend in home delivery was analyzed from PMA data”. Information that should be in the data section and not in the results.

Line 139: authors must justify the use of two statistical packages and what contribution to the study makes the analysis in two different software: STATA 17 and SPSS 25.

Line 141: it is redundant to point out that the logistic regression is binary. If applicable, mention whether the response variable had more than two categories so it was chosen to transform it into a binary.

Line 148-150. “For two regions (namely Tigray and Afar, two years (2022 and 2023 were missed.” Explain in limitations of the study, at the end of discussion, how did the loss of data affect the study?

The topics of discussion should come from the results of the study. The following is from the literature review and the authors did not raise it in the introduction: Line 178-184. The authors argue that births should be attended by personnel trained to detect and manage complications that may arise during childbirth. Do the authors assume that the personnel attending to deliveries at the women’s home are not sufficiently trained to detect and manage complications? Missing references. Explain in Introduction that home delivery care can become complicated if the attending midwife or obstetrician is insufficiently trained.

Line 188- 192. In 2019, a study reported the prevalence of home birth care at 28% vs. 37.80% found by the authors. To delve into the reasons for this difference of 9% between its finding and that reported by other authors previously.

Line 230. To name the regions. How does the underestimation of prevalence in these regions affect or not the trend identified in the country from 2019 to 2023?

Typo:

Line 178. “world health organization” is not correct. Most use capital letters.

Edit references with the style requested by PLOSOne.

Tables and figures contain insufficient information to understand the data they display.

Reviewers' comments:

Reviewer's Responses to Questions

**Comments to the Author**

1. Is the manuscript technically sound, and do the data support the conclusions?

Reviewer #1: Partly

Reviewer #2: Partly

2. Has the statistical analysis been performed appropriately and rigorously? 

Reviewer #1: No

Reviewer #2: Yes

3. Have the authors made all data underlying the findings in their manuscript fully available?

Reviewer #1: Yes

Reviewer #2: No

4. Is the manuscript presented in an intelligible fashion and written in standard English?

Reviewer #1: No

Reviewer #2: No

5. Review Comments to the Author

Reviewer #1: Thank you for inviting me to review the manuscript “Trend of Home Delivery and Its Associated Factors in Ethiopia during COVID 19 and Social Crisis (2019-2023)”. This paper needs English editing before having the potential to be published. In addition, a number of issues have to be addressed.

1) Abstract: method should be revised by detailing the study design, sample, sample size, data collection, etc.

2) Method: The instrument for outcome measure (Home Delivery/ Health facility delivery) should be clearly described.

3) Introduction and methods: The selection of “factors” and their measurement tools should be clearly stated. There are many factors influencing “place of delivery” potentially. The authors should explained why they selected the ones in this paper but not others. The instruments for measuring the factors should also be detailed.

4) Method: Did the authors combine four years data and then conduct the regression analyses?. Information on how the regression analyses were performed should be expanded.

5) Result: Standard deviation should be illustrated in figure 1.

6) Discussion: Strength of the study should be discussed

Reviewer #2: Thank you for giving me the opportunity to review this interesting manuscript. Measure against high mortality rates in Ethiopia and birth attendance of skilled health professional are very important topics. However, the manuscript needs thorough revision before publication could be considered. Especially the language needs to be improved. There are many sentences which are difficult to understand. Additionally, it would be good to replace the word “delivery” with “birth” which is a more active term and seems more appropriate. I would also suggest taking a more differentiated view on the problems around giving birth in Ethiopia and making a clearer distinction between the problem of home births, the circumstances in which they take place and the problem of births without skilled attendants.

Abstract

- Background, first sentences: mortality rates are not high in every part of the world, this first sentence seems to be too general.

- Background: I already wondered at this point if home births are the only problem or if their circumstances and that they take place without skilled health professionals is just as important.

- Objectives: I do not understand the “2024” in the end. In the title it was 2019-2023…

- Methods: were the data sets appended or merged?

- Results: “…showed a significant decline decreased from…” seems not to be correct. Additionally, “didn’t” is not academic writing.

Introduction

The introduction is logically structured and justifies the study well.

- The first sentence needs a reference.

- Line 51: is the information within the brackets correct?

- Lines 57-60: references are needed.

- Line 64: Reference 4 is not a reference by the World Health Organisation but from the NHS in the UK. Additionally, this reference takes a very differentiated view of the problem of place of birth. Under certain circumstances, home births can be recommended. However, the problem in Ethiopia is that women with high-risk pregnancies also give birth at home and are inadequately supported there.

- Line 67: Ethiopia is a member of what?

- Lines 69-72: this sentence is not correct and can not be understood.

- Line 74: the abbreviation DHS needs to be introduced correctly. This is true for all abbreviations, e.g. also ANC on line 82.

- Lines 87 and following: there is a confusion with the verb tenses in this paragraph.

Methods

- At the beginning of the chapter, the study design is missing, and it was also not mentioned at the beginning of the chapter Study Design and Data Source.

- Line 98: it needs to be explained, why the study was conducted in predominantly agrarian regions and this needs to be considered in the interpretation of the results.

Results

- Line 149: it should be “were missing” and in the next sentence “ranged” instead of “were ranged”. Please consider careful proofreading for this manuscript.

- Line 154 and other parts of the manuscript: inconsistencies in the number of decimals should be harmonized.

- Lines 155-156 as well as Table 2: the subdivision into the groups with overlapping years is very confusing and need explanations and justifications.

- Lines 169 und 172: I wonder if “counter parts” and “counterparts” are correct terms.

Discussion

The discussion needs some work. The first paragraph in the current version is not study related, misses references and would rather belong to the introduction. The first sentence of the discussion should include what was special for this study (if applicable) and/or what the authors intended to do. It should be followed by a very short, to the point summary.

- In my opinion, it should also be discussed and differentiated whether the problem is about home birth as such or about the circumstances in which home births take place (no risk assessment, probably no skilled health professionals, …).

- Lines 188-189: there is an inconsistency in the number of decimals.

- Lines 193-194: no new paragraph would be necessary as it continues with the same topic.

- Line 202: “avail” might not be the correct term.

- Lines 207-208 and line 221: the current study can support results from previous ones but not vis versa, because other study already existed before.

- Line 232: is it for sure an underestimation or is also an overestimation possible?

- Line 232: it probably should be “encouraging”.

Tables

- Table 1: Amhara 2019: a bracket is missing. Addis Ababa 2023: the space between the number and the bracket is missing.

- Table 2: as said above, the subdivision into these groups seem unclear with the overlapping years.

- Table 3: spaces between numbers and brackets are missing and there are inconsistencies with the number of decimals.

6. PLOS authors have the option to publish the peer review history of their article (what does this mean? ). If published, this will include your full peer review and any attached files.

**Do you want your identity to be public for this peer review?** For information about this choice, including consent withdrawal, please see our Privacy Policy .

Reviewer #1: No

Reviewer #2: **Yes: ** Susanne Grylka-Baeschlin

---

## [Author Response · Author response to Decision Letter 1]

13 Sep 2024

Rebuttal letter

From: Fekade Demeke Bayou (Department of Epidemiology and Biostatistics, School of Public Health, Wollo University, Dessie, Ethiopia)

Dear Editor, we are thankful for your giving us the chance to improve our manuscript. This rebuttal letter is prepared to address all comments and questions from the editor and reviewers. Thank you again.

1. Editing tables and figures, references and some writing errors.

Authors’ response: thank you for your concern. We have edited tables, figures and references according to the comments given by you and the reviewers.

2. Consistency between the different sections: for example the role of personnel who attend home births versus hospital births.

Authors’ response: thank you again. We specified that home births are commonly attended by traditional birth attendants, no one, and family or friend, who are considered unskilled to manage complications during and after birth (line 82-84). Similarly health facility birth is attended by skilled personnel including doctors, nurses/midwives, health officers, and health extension workers (line 101-102).

3. The reviewers agree on the need to unify the way of referring to home birth: birth or delivery

Authors’ response: thank you again. We addressed this issue by using the term “birth” instead of “delivery” consistently.

Additional editor’s comments

1. Abstract: the surveys analysed correspond to the period 2021-2023. However, the study period was from 2019 to 2023. Clarify this inconsistency

Authors’ response: thank you for your concern. Your concern is right. We used the data extraction period as the study period, this was reason for this discrepancy occurred. Now, we used the actual data collection period (2019-2023) as a study period (line 30 of the revised manuscript).

2. Line 70-71”… substantial prevalence of home birth preference in Ethiopia…” The ratio of births at home to hospital is missing. How much substantial?

Authors’ response: we mentioned this objectively on line #73 (73% of women give birth at home).

3. Line 74. Improve drafting. Clarify: Prevalence of which country Ethiopia or other African countries? The baseline should be reviewed to determine where the prevalence of 94.20% is from

Authors’ response: We made this statement more clear (line 74-76 of the revised manuscript).

4. Lack information about the association, if there is between home-assisted delivery and the prevalence of maternal morbidity and mortality due to bleeding, infection and high blood pressure…

Authors’ response: we added few points regarding the association of home delivery and maternal adverse health outcomes (line 85-88 of the revised manuscript).

5. Line 81-86…. Please clarify the idea and put separately the causes attributable to pregnant women and home birth, as well as the causes attributable to the health system, And human resources with insufficient training for childbirth care.

Authors’ response: Thank you for this comment again. We have tried to elaborate the major attributes of home delivery from pregnant women, family/house hold and health system dimensions (line 89-9).

6. Line 92: Check if the idea is correct. If the number of home births was not measured, why are you planning to expand services from 50% to 76%?

Authors’ response: We have checked again about these figures, the proportions of home delivery have been reported by Ethiopian DHS reports and a Systematic review and Meta-analysis study and the figures are correct. We have cited references (line 76-78).

7. Line 106-107: The date of the data source 2021-2023 does not coincide with the study period mentioned in the manuscript title: 2019-2023. Where the data for the years 2019 and 2020 come from.

Authors’ response: Thank you. We resolved this issue by using consistent study period (2019-2023) (line 107, 122).

8. Line 109-112. Authors should specify in detail the agency responsible for PMA surveys. Authors’ response: Thank you. The details of responsible institutions is clearly stated on line 127-134 of the revised manuscript.

9. Line 121-122. Reference of the sample frame missing: “Enumeration areas (EAs) were selected from the master sample frame of the Central Statistical Agency”.

Authors’ response: Thank you for this comment. We cited the reference for sample frame source on line 144.

10. Line 128-129: It is not clear how household surveys are included in the PMA.

Authors’ response: Thank you. We tried to elaborate the PMA household surveys as follow. The PMA conducted baseline surveys at household level and appointed eligible women from these surveyed households for the next surveys. In the prospective surveys, data were collected from appointed pregnant women resided in those households (from which baseline data were obtained). Hence, the PMA datasets contain both base line data collected at household level and consecutive survey data.

11. Line 100-104. Information to include in the introduction.

Authors’ response: Thank you. We moved the statement you mentioned to the introduction section (line 108 -111 of the revised manuscript).

12. Line 137-138: Information to be in the data source section

Authors’ response: Thank you. We moved the statement to the data source section (line 122-124 of the revised manuscript)

13. Line 147. “Four years trend in home delivery was analyzed from PMA data”. Information that should be in the data section and not in the results. Authors’ response: Thank you. We moved this statement to Data processing and analysis section (line 166-167 of the revised manuscript)

14. Line 139: authors must justify the use of two statistical packages and what contribution to the study makes the analysis in two different software: STATA 17 and SPSS 25.

Authors’ response: Thank you. This might have no special contribution to the study. Using multiple software may not guarantee data quality. However, since we used STATA to append the datasets and SPSS for all further analysis, we believe that being genuine to state what we did is a good behavior in scientific writing.

15. Line 141: it is redundant to point out that the logistic regression is binary. If applicable, mention whether the response variable had more than two categories so it was chosen to transform it into a binary.

Authors’ response: Thank you. We addressed this comment by removing the redundancy (line 170 of the revised manuscript). Just nothing was transformed as the outcome variable (place of birth) was a dichotomous variable (home birth Vs health facility birth).

16. Line 148-150. “For two regions (namely Tigray and Afar, two years (2022 and 2023 were missed.” Explain in limitations of the study, at the end of discussion, how did the loss of data affect the study? Authors’ response: Thank you for this critical insight. We stated suggested effect of this missing information under limitation of the study section (line 261-264).

17. Line 178-184 …Explain in Introduction that home delivery care can become complicated if the attending midwife or obstetrician is insufficiently trained

Authors’ response: Thank you, In Ethiopia, home birth is usually attended by unskilled individuals including traditional birth attendants, family members and friends, even by no one (line 82-84). It is easy to predict what will happen if births are attended by unskilled individuals at home (line 85-88).

18. Line 188- 192. In 2019, a study reported the prevalence of home birth care at 28% vs. 37.80% found by the authors. To delve into the reasons for this difference of 9% between its finding and that reported by other authors previously. Authors’ response: Thank you for this brilliant insight: We were happy if we could do that. To compare differences of proportions, we tried to search similar studies (trend analysis) which assessed changes in the prevalence/proportion of home birth across different times. However, as far as we searched, we couldn’t find any study which reported differences. As a result, we discussed the latest figure (in 2023) with other studies’ findings. We are open minded to accept if you suggest any better way to discuss our finding.

19. Line 230. To name the regions. How does the underestimation of prevalence in these regions affect or not the trend identified in the country from 2019 to 2023?

Authors’ response: Thank you: in fact, missing data may cause under or over estimation of the overall figure of home birth in the country. Both of which causes biased conclusion. Underestimation of the overall figure of home birth in these dominant Ethiopia regions may mask the real condition of the problem. As a result this biased information may mislead further decision making processes.

20. Line 178. “World health organization” is not correct. Most use capital letters.

Authors’ response: Thank you, we addressed this issue by capitalizing each cases everywhere in the revised manuscript.

21. Edit references with the style requested by PLOSOne.

Authors’ response: We tried to follow the Vancouver style for our reference citation.

22. Tables and figures contain insufficient information to understand the data they display. Authors’ response: we tried to add relevant information in all tables and figure.

Reviewer #1:

We would like to forward our deepest gratitude to the reviewer for constructive comments and scientific insights. We have tried to address comments and prepared the following point by point response.

1) . This paper needs English editing before having the potential to be published.

Authors’ response: Thank you, we strived to improve the language issues by consulting English language experts.

Abstract:

2) Method should be revised by detailing the study design, sample, sample size, data collection, etc.

Authors’ response: Thank you for these helpful insights. We added the study design and sample on line 30-31 of the revised manuscript. Regarding data collection, since we used secondary data collected by PMA Ethiopia project, we focused on how the dataset were obtained from the repository and processed for analysis. We are open minded to accept any direction you will give to us.

Method:

3) The instrument for outcome measure (Home Delivery/ Health facility delivery) should be clearly described.

Authors’ response: Thank you. We clearly stated how the outcome was assessed on 158-164 of the revised manuscript.

4) Introduction and methods: The selection of “factors” and their measurement tools should be clearly stated. There are many factors influencing “place of delivery” potentially. The authors should explained why they selected the ones in this paper but not others. The instruments for measuring the factors should also be detailed.

Authors’ response: Thank you the project managers used interviewer administered questionnaire as a data collection tool. Selection of factors was based on preliminary analysis (Bivariable logistic regression analysis), and expertise knowledge on those variables. We stated on line 172-174 of the revised manuscript. However, since we used secondary variables, we missed some variables which may potentially affect place of birth. This might be disclosed as the limitation of this study.

5) Method: Did the authors combine four years data and then conduct the regression analyses? Information on how the regression analyses were performed should be expanded.

Authors’ response: we didn’t merged the four years data to run regression analysis. Instead we used the latest year (2023) data to rum logistic regression. Because we believe that identifying factors which affect the recent trend is more important and logical to forward recommendations.

6) Result: Standard deviation should be illustrated in figure 1.

Authors’ response: Thank you for your direction. We added the standard deviation error bars on Figure 1

7) Discussion: Strength of the study should be discussed

Authors’ response: we mentioned the strength of the study on line 257-259 of the revised manuscript.

Reviewer 2

We would like to forward our deepest gratitude for giving us these constructive comments and scientific insights. We have tried to address comments and prepared the following point by point response.

1. … The language needs to be improved.

Authors’ response: Thank you, we strived to improve the language issues by consulting English language experts.

2. It would be good to replace the word “delivery” with “birth” which is a more active term and seems more appropriate.

Authors’ response: Thank you for your insight. We replaced the word “delivery” with “birth” throughout the document.

3. I would also suggest taking a more differentiated view on the problems around giving birth in Ethiopia and making a clearer distinction between the problem of home births, the circumstances in which they take place and the problem of births without skilled attendants.

Authors’ response: Thank you, In Ethiopia, home births are usually attended by unskilled individuals including traditional birth attendants, family members and friends, even by no one (line 82-84). It is easy to predict what will happen if births are attended by unskilled individuals at home (line 85-88).

Abstract –

4. Background, first sentences: mortality rates are not high in every part of the world, this first sentence seems to be too general.

Authors’ response: Thank you for this comment. We corrected this statement, line 21-22 of the revised manuscript.

5. Background: I already wondered at this point if home births are the only problem or if their circumstances and that they take place without skilled health professionals is just as important.

Authors’ response: Thank you again. In Ethiopian condition, as we stated above, what makes home births bad is that attended by unskilled individuals (line 82-84).

6. Objectives: I do not understand the “2024” in the end. In the title it was 2019-2023…

Authors’ response: Thank you for this comment. The year “2024” refers the data retrieval (obtained from the PMA data repository) year while 2019-2023 refers the data collection period. We understand this statement may cause confusion among readers. Now, we corrected the year in line with the title (line 28 of the revised manuscript).

7. Methods: were the data sets appended or merged?

Authors’ response: The datasets were appended to make it complete. The PMA datasets have baseline survey data as well as prospective follow up survey data. We appended them to get complete datasets.

8. Results: “…showed a significant decline decreased from…” seems not to be correct. Additionally, “didn’t” is not academic writing.

Authors’ response: Thank you again, we made corrections on these errors (line 37 and 40 of the revised manuscript).

Introduction

9. The first sentence needs a reference.

Authors’ response: Thank you for this concern, we cited references (line 58).

10. - Line 51: is the information within the brackets correct?

Authors’ response: Yes, it was reported by the World Health Organization (WHO).

11. - Lines 57-60: references are needed. Authors’ response: Thank you, we cited references (line 65-67).

12. Line 64: Reference 4 is not a reference by the World Health Organisation but from the NHS in the UK. …However, the problem in Ethiopia is that women with high-risk pregnancies also give birth at home and are inadequately supported there. Authors’ response: Thank you for this concern. We corrected the reference. The context of home birth is completely different from that of recommended in some countries. We clearly stated this context on line 82-84 of the revised manuscript.

13. Line 67: Ethiopia is a member of what?

Authors’ response: Thank you, we improved this statement (line 70). We mean Ethiopia is one of the countries committed to meet SDG agenda 3.1.

14. - Lines 69-72: this sentence is not correct and cannot be understood.

Authors’ response: We tried to clarify this statement (line 73-75 of the revised manuscript).

15. - Line 74: the abbreviation DHS needs to be introduced correctly. This is true for all abbreviations, e.g. also ANC on line 82.

Authors’ response: We addressed this issue by stating the full word of abbreviations for the first time (line 75, 194).

16. - Lines 87 and following: there is a confusion with the verb tenses in this paragraph. Authors’ response: Thank you again, we improved the gramma

---

## [Editor Report · Decision Letter 1]

1 Oct 2024

PONE-D-24-10835R1Trend of Home Birth and Its Associated Factors in Ethiopia during COVID-19 and Social Crisis (2019-2023)PLOS ONE

Dear Dr. Bayou,

Thank you for submitting your manuscript to PLOS ONE. After careful consideration, we feel that it has merit but does not fully meet PLOS ONE’s publication criteria as it currently stands. Therefore, we invite you to submit a revised version of the manuscript that addresses the points raised during the review process.

**ACADEMIC EDITOR: **The manuscript must be edited, particularly the tables lack of title and source.

Please submit your revised manuscript by Nov 15 2024 11:59PM. If you will need more time than this to complete your revisions, please reply to this message or contact the journal office at plosone@plos.org . Please include the following items when submitting your revised manuscript:

A marked-up copy of your manuscript that highlights changes made to the original version. You should upload this as a separate file labeled 'Revised Manuscript with Track Changes'.An unmarked version of your revised paper without tracked changes. You should upload this as a separate file labeled 'Manuscript'.

We look forward to receiving your revised manuscript.

Kind regards,

Doris Verónica Ortega-Altamirano, PhD

Academic Editor

PLOS ONE

Journal Requirements:

Additional Editor Comments :

The manuscript is suitable for publication in the journal.

Tables lack of titles and sources.

Table 1. Discriptions of the study participants by region and survey years in Ethiopia, 2024 (the study is from 2019 to 2023. The year in the table´s title is wrong.

Table 2. Percent change in prevalence of home delivery from 2019-2023, Ethiopia, 2024. The same that above.

Table 3 Multivariable logistic regression analysis of the factors associated with home birth in Ethiopia, 2024. The same comment in table 1 and 2.
---

## [Author Response · Author response to Decision Letter 2]

7 Oct 2024

Dear Dr Doris Veronica Ortega-Altamirano, I would like to forward my heart felt regards for

your commitment to handle the review processes of our manuscript. Your timely feedback

and helpful recommendations are also highly admired. We addressed your comments as

follows:

1. Journal requirement: please review your reference list to ensure that it is complete and

correct…

Authors’ response: Thank you for your concerns. We checked for completeness of

references’ citation and we made some corrections (reference #4, and 8). We ensured that we

didn’t use retracted any reference. Additional comments

1. Tables lack of titles and sources

- Table 1 … the year in the table’s title is wrong

Authors’ response:Thank you, we addressed this comment by changing the year as 2019- 2023 and adding source of data i.e PMA Ethiopia (line 185-186 of the revised manuscript)

-Table 2 and 3, the same comment above. We also addressed similar comments on table 2 and

3 (line 198-199 and 217-218 respectively)

NB. We didn’t get attachments of reviewers’ comments in this round.

---

## [Decision Letter · Decision Letter 2]

20 Nov 2024

PONE-D-24-10835R2Trend of Home Birth and Its Associated Factors in Ethiopia during COVID-19 and Social Crisis (2019-2023)PLOS ONE

Dear Dr. Bayou,

Thank you for submitting your manuscript to PLOS ONE. After careful consideration, we feel that it has merit but does not fully meet PLOS ONE’s publication criteria as it currently stands. Therefore, we invite you to submit a revised version of the manuscript that addresses the points raised during the review process. The manuscript addresses a topic of interest but does not explain it sufficiently and clearly.It presents multiple areas for improvement and requires authors to respond to reviewers' comments. Please submit your revised manuscript by Jan 04 2025 11:59PM. If you will need more time than this to complete your revisions, please reply to this message or contact the journal office at plosone@plos.org . Please include the following items when submitting your revised manuscript:

We look forward to receiving your revised manuscript.

Kind regards,

Doris Verónica Ortega-Altamirano, PhD

Academic Editor

PLOS ONE

Additional Editor Comments:

The manuscript addresses a topic of interest but does not explain it sufficiently and clearly. Presents multiple areas for improvement and requires authors to respond to reviewers' comments.

Reviewers' comments:

Reviewer's Responses to Questions

**Comments to the Author**

1. If the authors have adequately addressed your comments raised in a previous round of review and you feel that this manuscript is now acceptable for publication, you may indicate that here to bypass the “Comments to the Author” section, enter your conflict of interest statement in the “Confidential to Editor” section, and submit your "Accept" recommendation.

Reviewer #2: All comments have been addressed

Reviewer #3: All comments have been addressed

2. Is the manuscript technically sound, and do the data support the conclusions?

Reviewer #2: Yes

Reviewer #3: Yes

3. Has the statistical analysis been performed appropriately and rigorously? 

Reviewer #2: Yes

Reviewer #3: Yes

4. Have the authors made all data underlying the findings in their manuscript fully available?

Reviewer #2: Yes

Reviewer #3: Yes

5. Is the manuscript presented in an intelligible fashion and written in standard English?

Reviewer #2: Yes

Reviewer #3: Yes

6. Review Comments to the Author

Reviewer #2: Thank you for thoroughly revising the manuscript. I evaluated R1 and R2 together and the manuscript improved considerably. There are still a few minor comments:

- Thank you for harmonising the decimals. While this is necessary and makes sense in the result section, it is not adequate in the introduction in formulations such as “nearly 95%” and “around 70%”. Please remove the decimals there.

- Lines 108-110: word repetition of “population”, which should be avoided.

- Title “Limitation of the study” should now be “Strength and limitation of the study”

- Line 264: I do not understand the numbers in the bracket (2022 and 149 2023)

- Line 265: The term “probably” does not make sense in this context. You could write: “… might have under- or overestimated the proportion…”

Reviewer #3: I am very grateful to be part of this peer review; it is a pleasure to read about such an important topic for women’s health and well-being as the trend of home births in Ethiopia and its associated factors. Below, I will outline some key points to strengthen the rigor and clarity of the manuscript.

In the introduction section, specifically in line 91, the influence of cultural beliefs and social norms on the preference for home births is not addressed. Incorporating these factors, could provide a more comprehensive understanding of the cultural and social factors influencing the choice of birth location.

In line 118, it is mentioned that the sampling was conducted only in certain regions of Ethiopia, which is inconsistent with the conclusions that state the results are representative of the entire country.

In line 141, the two-stage cluster sampling technique is described. I suggest providing a more detailed description of the sampling process at each stage, including the criteria for selecting clusters and study units.

Finally, in line 154, variables such as awareness, perception, knowledge, and use of contraceptive methods, components of health service delivery, perceived quality, and side effects among current users, birth history, fertility intentions, and empowerment domains are mentioned. However, the absence of variables such as educational level, family income, and geographical location is significant, as they are essential for controlling potential confounding factors, which could enrich the study’s conclusions.

7. PLOS authors have the option to publish the peer review history of their article (what does this mean? ). If published, this will include your full peer review and any attached files.

**Do you want your identity to be public for this peer review?** For information about this choice, including consent withdrawal, please see our Privacy Policy .

Reviewer #2: **Yes: ** Susanne Grylka-Baeschlin

Reviewer #3: **Yes: ** PhDc Gustavo Alejandro Román-Brito

---

## [Author Response · Author response to Decision Letter 3]

6 Dec 2024

Rebuttal Letter

PONE-D-24-10835R2

Trend of Home Birth and Its Associated Factors in Ethiopia during COVID-19 and Social Crisis (2019-2023)

From: Fekade Demeke Bayou (Department of Epidemiology and Biostatistics, School of Public Health, Wollo University, Dessie, Ethiopia)

Dear Editor, we would like to forward our heartfelt thanks for your commitment and timely feedback during the review process of this manuscript. We are also thankful to the reviewers for investing their time and energy on this manuscript. All concerns from reviewers and editor are important to improve the readability of this manuscript, hence, we are satisfied by your service. This point by point response letter is prepared to address all comments and questions from the reviewers. Thank you again.

Reviewer 2

1. ….In the introduction in formulations such as “nearly 95%” and “around 70%”. Please remove the decimals there

Authors’ response: thank you for your concern. We removed the decimals.

2. Lines 108-110: word repetition of “population”, which should be avoided.

Authors’ response: Thank you we removed the redundancy (line 113 of the revised manuscript)

3. Title “Limitation of the study” should now be “Strength and limitation of the study”

Authors’ response: thank you for your concern, we changed the subsection as “Strength and limitation of the study” on line 260 of the revised manuscript.

4. Line 264: I do not understand the numbers in the bracket (2022 and 149 2023)

Authors’ response: Thank you so much! It was an editorial error. We corrected it now, line 266 of the revised manuscript.

5. Line 265: The term “probably” does not make sense in this context. You could write: “… might have under- or overestimated the proportion…”

Authors’ response: Thank you again, we accepted your comments and corrected as indicated on line 267 of the revised manuscript.

Reviewer #3:

1. In the introduction section, specifically in line 91, the influence of cultural beliefs and social norms on the preference for home births is not addressed. Incorporating these factors

Authors’ response: Thank you so much! This is crucial information missing in our manuscript. Now, we added such evidences on line 93-97 of the revised manuscript.

2. In line 118, it is mentioned that the sampling was conducted only in certain regions of Ethiopia, which is inconsistent with the conclusions that state the results are representative of the entire country.

Authors’ response: Thank you. This is critical concern. Even though the samples were not taken from every region in the country, due to multi-sage sampling, representativeness is more likely since the sample of regions were selected from each stratum (urban and rural). More over the number of regions (more than 50% of all region in the country were included) included in the study were adequate to represent the national figure of home birth. Hence, the sampling doesn’t compromise representativeness while using statistically efficient sampling method. However, doing all these, cannot assure complete representativeness, for which we declare limitation of the study on line 262-269 of the revised manuscript.

3. In line 141, the two-stage cluster sampling technique is described. I suggest providing a more detailed description of the sampling process at each stage, including the criteria for selecting clusters and study units.

Authors’ response: Thank you for your request for clarification. We tried to clarify the stages in sampling procedure. As we stated on line 142-148 of the revised manuscript, two stage sampling was:

Stage 1: selection of Enumeration areas from the two strata (urban and rural).

Stage 2: selection of households from the selected enumeration areas.

4. Finally, in line 154, variables such as awareness, perception, knowledge, and use of contraceptive methods, components of health service delivery, perceived quality, and side effects among current users, birth history, fertility intentions, and empowerment domains are mentioned. However, the absence of variables such as educational level, family income, and geographical location is significant, as they are essential for controlling potential confounding factors, which could enrich the study’s conclusions

Authors’ response: Thank you again. Your concern is correct. However, as we used secondary data, such important variables were missing from the dataset.

---

## [Decision Letter · Decision Letter 3]

26 Feb 2025

Trend of Home Birth and Its Associated Factors in Ethiopia during COVID-19 and Social Crisis (2019-2023)

PONE-D-24-10835R3

Dear Dr. Bayou,

We’re pleased to inform you that your manuscript has been judged scientifically suitable for publication and will be formally accepted for publication once it meets all outstanding technical requirements.

Kind regards,

Tegene Atamenta Kitaw (MSc, MPH), Woldia University

Academic Editor

PLOS ONE

Additional Editor Comments (optional):

Reviewers' comments:

Reviewer's Responses to Questions

**Comments to the Author**

1. If the authors have adequately addressed your comments raised in a previous round of review and you feel that this manuscript is now acceptable for publication, you may indicate that here to bypass the “Comments to the Author” section, enter your conflict of interest statement in the “Confidential to Editor” section, and submit your "Accept" recommendation.

Reviewer #2: All comments have been addressed

2. Is the manuscript technically sound, and do the data support the conclusions?

Reviewer #2: Yes

3. Has the statistical analysis been performed appropriately and rigorously? 

Reviewer #2: Yes

4. Have the authors made all data underlying the findings in their manuscript fully available?

Reviewer #2: Yes

5. Is the manuscript presented in an intelligible fashion and written in standard English?

Reviewer #2: Yes

6. Review Comments to the Author

Reviewer #2: (No Response)

7. PLOS authors have the option to publish the peer review history of their article (what does this mean? ). If published, this will include your full peer review and any attached files.

**Do you want your identity to be public for this peer review?** For information about this choice, including consent withdrawal, please see our Privacy Policy .

Reviewer #2: **Yes: ** Susanne Grylka-Baeschlin

---

## [Editor Report · Acceptance letter]

PONE-D-24-10835R3

PLOS ONE

Dear Dr. Bayou,

I'm pleased to inform you that your manuscript has been deemed suitable for publication in PLOS ONE. Congratulations! Your manuscript is now being handed over to our production team.

Kind regards,

on behalf of

Dr. Tegene Atamenta Kitaw

Academic Editor

PLOS ONE